# OpenReview forum: "Inference-Based Privacy Violations Demand Immediate Recognition as a Distinct AI Safety Priority"
_NeurIPS.cc/2025/Position_Paper_Track — Submitted to NeurIPS 2025 Position Paper Track_

### Official Review · Reviewer_vzZf · 2025-08-11

**Significance:** 3
**Presentation:** 3
**Rating:** 4
**Confidence:** 4

**Summary:**

This position paper argues that inference-based privacy (IBP) risks should be recognized as a distinct and urgent AI safety priority. This paper defines IBP risks as the capacity of AI systems to deduce sensitive personal information from seemingly harmless data, a threat they contend is inadequately addressed by current technical and regulatory frameworks.

**Strengths:**

S1: The paper excels at defining Inference-Based Privacy (IBP) risk and distinguishing it from traditional data privacy issues like data breaches. It formally defines an IBP violation as a model inferring a sensitive attributes from non-sensitive data z about an individual who was not in the training dataset, which clarifies why methods focused on data access are insufficient. This distinction is crucial because it highlights that the privacy harm occurs from the generation of new, sensitive insights, not the leakage of existing data.


S2: he paper provides a thorough critique of why existing privacy protection measures are inadequate for addressing IBP risks. It systematically dismantles the effectiveness of technical solutions like differential privacy and federated learning, explaining their limitations in preventing inferences based on population-level patterns. Furthermore, it identifies specific gaps in major regulations like the GDPR and the EU AI Act, which are not designed to handle passive inference that doesn't lead to immediate, significant effects.

**Weaknesses:**

O1: The paper's technical solutions, like "Inference Detection," may understate their implementation difficulty. It acknowledges challenges are "genuine" but is overly optimistic, as detecting undefined "inference overreach" is a significant leap compared to other AI safety issues, making the solutions seem more feasible than they are.

O2: The rebuttal to the "Innovation and Utility Argument" is underdeveloped. It suggests consent as a solution but doesn't fully grapple with the powerful economic incentives driving these capabilities, failing to explore the market forces that resist privacy constraints.

O3: Using AlphaEvolve as a precedent for escalating risk relies on speculation about "spontaneous emergence." By calling it a "plausible risk," the paper shifts from its evidence-based arguments to a less certain claim, which detracts from its otherwise strong grounding in documented phenomena.

**Questions:**

N/A

**Alternative Position:**

Yes, and alternative positions are well-considered and addressed by the argument

**Author Identification:**

No.

**Context:**

4

**Discussion:**

4

**Ethics:**

["NO or VERY MINOR ethics concerns only"]

**Position:**

Yes, the paper argues for or against a position related to machine learning.

**Support:**

4

**Thoroughness:**

4

---

### Official Review · Reviewer_NFDT · 2025-08-16

**Significance:** 4
**Presentation:** 4
**Rating:** 7
**Confidence:** 4

**Summary:**

The paper argues that Inference-Based Privacy (IBP) risks, where AI systems generate sensitive information through inference rather than exposing stored data, are an under-recognized but critical category of AI safety risk. The paper calls on policymakers, researchers, and practitioners to treat IBP as a distinct area requiring urgent attention.

The paper starts by formally defining IBP risk and distinguishing IBP risk from other AI privacy risks because it involves generating novel sensitive data through pattern recognition, not retrieving stored data. The author(s) provide illustrative examples of this risk from recent research. The paper goes on to describe why current privacy risk mitigations fall short in addressing IBP risk. The paper also highlights gaps in current regulatory privacy frameworks to protect individuals against IBP risk.

The paper propose a framework for addressing IBP risk, including technical and regulatory approaches (constraining inference capabilities as part of AI safety) and research priorities. The author(s) address potential objections to their arguments and end with a call to action to address IBP risk in this "critical window for action."

**Strengths:**

This paper is very well written, with a clear argument. It makes a compelling case for focus on an underrepresented area of AI safety research. The definition of IBP risk and its distinction from other types of privacy risks is well supported by literature illustrating the occurrence of IBP risk across a variety of domains.

The paper flows logically from this theoretical definition to clearly articulating why current approaches to privacy risk mitigation are insufficient for mitigating IBP risk. The framework that the author(s) propose to address these gaps is systematic and internally consistent, providing a solid foundation for future research.

The author(s) have also done an excellent job of anticipating potential objections to their argument and addressing each of these objections to make their argument more compelling.

**Weaknesses:**

The paper's description of current regulatory frameworks and their gaps is lacking in supporting evidence or literature, with only brief references to specific regulations (GDPR, CCPA) and insufficient specificity about their regulatory gaps. Additional detail would strengthen the author(s)' arguments that regulatory frameworks must evolve to meaningfully address IBP risk.

Some arguments are repeated in the conclusion, making the contribution feel less concise than it could be and resulting in the reader feeling that the argument has been over-argued.

The paper acknowledges but underplays the potential negative impact on innovation of constraining inference, which may weaken the persuasiveness of the policy argument.

**Questions:**

Your framework emphasizes constraining inference capabilities to reduce IBP risk. How do you envision balancing this with legitimate use cases where inference is essential (e.g., clinical decision support, fraud detection)? Could you elaborate on how to distinguish between harmful vs. beneficial inferences in a way that can be operationalized during model design or auditing?

**Alternative Position:**

Yes, and alternative positions are well-considered and addressed by the argument

**Author Identification:**

No.

**Context:**

3

**Discussion:**

4

**Ethics:**

["NO or VERY MINOR ethics concerns only"]

**Position:**

Yes, the paper argues for or against a position related to machine learning.

**Support:**

3

**Thoroughness:**

4

---

### Official Review · Reviewer_9MTw · 2025-08-20

**Significance:** 2
**Presentation:** 1
**Rating:** 2
**Confidence:** 4

**Summary:**

This position paper argues that Inference-Based Privacy (IBP) which is the ability of AI systems to infer sensitive personal information from seemingly innocuous inputs, should be recognized as an urgent and distinct AI safety priority. The authors distinguish IBP from traditional data leakage, highlight concrete risks across domains such as medical imaging, large language models, vision-language models, and autonomously evolving systems, and emphasize the inadequacy of current technical and regulatory safeguards. To address these challenges, the paper proposes a multi-faceted framework spanning technical interventions, policy and regulatory measures, and new research priorities, ultimately calling for immediate collective action to prevent IBP violations from becoming entrenched in future AI systems.

**Strengths:**

1. The paper highlights an timely issue and argues for its recognition as a distinct AI safety priority, using concrete examples across domains (medical AI, LLMs, vision-language models).

2. The structure is coherent and the inclusion of alternative perspectives, making the topic relevant and potentially valuable for community discussion at NeurIPS.

**Weaknesses:**

1. The paper is very short (6.5 pages) relative to NeurIPS standards and relies solely on text, without any figures or tables to improve readability or to summarize comparisons for general readers.

2. While concrete examples are provided, the evidence base is limited: citations and descriptions are limited, there is no systematic summary of existing privacy-related techniques, and no comparative analysis (which could be illustrated with charts or tables).

3. Although it is a position paper, the contribution and position remain overly abstract, as the manuscript repeatedly emphasizes the urgency of the issue without offering substantive discussion of potential mitigation strategies. The proposed “Framework for Immediate Action” is presented only at a high level, lacking concrete potential methodological tools, design blueprints, or feasible solution pathways.

4. The discussion of alternative views is underdeveloped: All arguments are framed in generic terms (“critics,” “some might contend”) without any references or supporting evidence, weakening the credibility of the counter-argumentation.

**Questions:**

1. Could the authors provide a more systematic overview of existing privacy-related techniques (e.g., differential privacy, federated learning, secure computation) and clarify how their proposed framing of IBP uniquely extends or differs from prior work?

2. The proposed “Framework for Immediate Action” is described at a high level. Can the authors elaborate with possible methodological tools, design blueprints, or example pathways that could make this framework more actionable for the ML community?

3. The discussion of alternative views is framed in generic terms (“critics,” “some might contend”) without references. Could the authors cite and engage with specific prior works or stakeholders who hold these views, to strengthen the credibility of their counter-arguments?

4. Since the paper calls for privacy is inherently interdisciplinary, how do the authors envision integrating insights from some subjects such as law, social sciences, or HCI into the development of IBP mitigation strategies?

**Alternative Position:**

Yes, and alternative positions are well-considered and named but not addressed

**Author Identification:**

No.

**Context:**

1

**Discussion:**

3

**Ethics:**

["NO or VERY MINOR ethics concerns only"]

**Position:**

Yes, the paper argues for or against a position related to machine learning.

**Support:**

1

**Thoroughness:**

5

---

### Note · Authors · 2025-09-05

**1-10 Additional Comments:**

This survey format is unusual to us and it has made it difficult to format responses. It would be nice to have a clearer response and reviewer interaction expectation ahead of time for next year. This was a very enjoyable track for us to submit our work to.

**1-11 Submit Again:**

Unsure

**1-1 Submission Process:**

4

**1-2 Next Year:**

It would be nice to have several, quicker back and forth interactions with reviewers and authors next year before the final decisions.

**1-3 Future Development:**

There could be an interesting experiment to run where all the paper authors are required to participate in producing short, permanently anonymized reviews on 2 or 3 other position papers. They don't need to be thorough, but they would provide a space for early discussions and informal feedback to be shared. Although not everyone is looking to review more papers, this might establish a unique feel that guarantees discussions start to take place. It may also reduce the number of submissions.

**1-4 Interest:**

["Panel discussions with other position paper authors", "Structured debates on controversial topics"]

**1-5 Thoughtful:**

7

**1-6 Supportive:**

10

**1-7 Technical Aspects Versus Position:**

3

**1-8 Gate Keeping:**

3

**1-9 Camera Ready Changes:**

We will fix certain terminology usage and phrasing that created avoidable misinterpretations and streamline the conclusion to reduce redundancy while keeping the focus on the core position. We will also add a couple figures to improve clarity, though not the specific one a reviewer suggested as that will not show anything interesting. In addition, we will provide brief clarifications to acknowledge related considerations such as economic pressures and comparative methods, framing these as open discussion areas rather than overextending the current scope of the arguments. Finally, we will add clarifications at the start of the discussion of alternative views to make it clear that the novelty and neglected nature of the position require that we predict and react to the likely criticisms.

These changes should improve clarity and readability, address reviewer concerns, and reinforce the paper’s position without altering our core position at all. The last part is critical since all reviewers agree with the stated position.

**3-1 Review Response1:**

VzZf

**3-2 Reaction To Review1:**

We appreciate this reviewer’s thoughtful engagement and recognition of the paper’s core contributions. Their summary reflects a clear understanding of IBP risks as a distinct privacy concern and the limitations of current technical and regulatory safeguards. We especially value their articulation of how the paper distinguishes IBP from traditional privacy harms and their acknowledgment of the systematic critique of existing solutions.

Regarding the identified weaknesses (O1 and O3), we recognize that some terminology may have been presented in a way that allowed for misinterpretation. While our intended usage was consistent and supported by context, we understand how this could affect perception. These are straightforward issues to address, and we are grateful to the reviewer for highlighting them.

The second weakness (O2) raises an important point about extending the discussion to economic pressures, but this falls outside the scope of the current paper. Our argument was meant to establish a financially pressure-agnostic position that provides a framework separating powerful financial incentives from the harmful impacts of privacy violations by promoting a clear path toward continued utility with respect for privacy. We see this as foundational work that sets the stage for future discussions, including those suggested by the reviewer.

Overall, this review demonstrated deep and constructive engagement with the paper’s position and is supportive of it.

**3-3 Review Response2:**

NFDT

**3-4 Reaction To Review2:**

We greatly appreciate this reviewer’s recognition of the paper’s clarity, systematic argument, and significance. Their summary demonstrates a strong understanding of the position and its relevance to AI safety. The feedback is thoughtful and constructive, and we found the questions particularly helpful in identifying focal points for improvement.

The reviewer’s observation about the repetitiveness of the conclusion is especially valuable. We had anticipated the possibility that some readers might lose focus on the precise position throughout the paper, so we are pleased that this reviewer felt differently and provided actionable feedback. We will address this in the next revision.

Overall, this review was highly supportive of the position and concentrated on enhancing the paper’s clarity and completeness. We sincerely thank the reviewer for their genuine engagement and constructive input.

**3-5 Review Response3:**

9MTw

**3-6 Reaction To Review3:**

We appreciate this reviewer’s recognition of the timeliness of the topic and the coherent structure of our submission. Their summary accurately captures the core position and its relevance across domains. However, we find a notable disconnect between these positive observations and the final score assigned.

The low score suggests that the position, as presented, contains a substantial flaw or drawback, yet the listed weaknesses seem somewhat peripheral to the actual position. Some comments seemed less focused on the position itself, such as weakness 2, which calls for a comparative analysis of methods that are equally ineffective at addressing this problem, and weakness 3, which requests solution discussions even though a critique of the position would argue that IBP risks are neither an urgent AI safety priority nor meaningfully distinct from other AI safety issues. Additionally, the review appears to pitch the novelty of the position against itself by noting that the discussion of alternative views is underdeveloped due to a lack of cited critics. In reality, the absence of such critics reflects the reason this position is necessary: it highlights an undefined and neglected area that demands attention. Similarly, remarks on length and the absence of figures did not convey a strong engagement with the position.

That said, the review included valuable suggestions that we will incorporate in the next version, such as adding figures. We especially appreciated the final question, which may be the most insightful we received, as it engages directly with the position at the depth we believe is necessary and helps move the discussion forward. Given that the reviewer agreed with the position and posed such a constructive question, the assigned rating seems misaligned with their expressed understanding.

---

### Meta-Review · Area_Chair_ZkAP · 2025-09-12

**Rating:** 4
**Confidence:** 5

**Strengths:**

The reviewers praise the timely nature of the privacy risk presented in the paper and its capacity to raise interesting discussions, as well as distinctions made wrt other privacy risks.

**Weaknesses:**

The paper could be improved by:
- Enhancing readability with figures or tables to clearly summarize comparisons.
- Adding more references and empirical evidence to substantiate claims.
- Discussing the impact of the proposed solution on the quality of service in legitimate use cases, as well as technical challenges around query monitoring.
- Providing more detail on the gaps in current regulations. Specifically, some of existing regulatory frameworks already require data controllers to account for risks such as singling out, linkability, and inference. Although these regulations were established before the advent of GPAI systems, the core challenge lies less with the adequacy of the legal frameworks, and more with the burden on data controllers to demonstrate that the data they release cannot be exploited by GPAI models to infer sensitive information. Furthermore, existing regulations already contain provisions addressing inference; for example, GDPR includes explicit restrictions on profiling.
- The paper suggests a contradiction with DP, but by definition DP guarantees that, regardless of auxiliary information available to an attacker, the output of a differentially private computation does not increase risk.

**Questions:**

The reviewers raised thoughtful questions regarding the challenges of query monitoring and its potential impact on legitimate use cases. It would also be valuable to explore how to account for the availability of open-weight models and clarify how IPB will affect the responsibilities of data controllers.

**Ethics:**

No ethical violations or concerns were raised by the reviewers.

**Thoroughness:**

5

---

### Decision · Program_Chairs · 2025-09-26

Reject